# Prediction of the Binding to the Nuclear Factor NF-Kappa-B by Constituents from *Teucrium polium* L. Essential Oil

**DOI:** 10.3390/cimb47010048

**Published:** 2025-01-14

**Authors:** Renilson Castro de Barros, Renato Araújo da Costa, Nesrine Guenane, Boulanouar Bakchiche, Farouk Benaceur, Omer Elkiran, Suelem Daniella Pinho Farias, Vanessa Regina Silva Mota, Maria Fani Dolabela

**Affiliations:** 1Pharmaceutical Sciences Postgraduate Program, Federal University of Pará, Belém 66075-110, PA, Brazil; renilsonbarros098@gmail.com; 2Federal Institute of Education Sciences of the State of Pará, Campus Abaetetuba, Abaetetuba 68440-000, PA, Brazil; renato.costa@ifpa.edu.br; 3Laboratory of Biological and Agricultural Sciences (LBAS), Amar Telidji University, Laghouat 03000, Algeria; n.guenane.gpr@lagh-univ.dz (N.G.); f.benaceur@lagh-univ.dz (F.B.); 4Research Unit of Medicinal Plant (RUMP), Biotechnology Research Center (CRBt), Laghouat 03000, Algeria; 5Department of Environmental Health, Vocational School of Health Services, Sinop University, 57000 Sinop, Turkey; omer_elkiran@hotmail.com; 6Federal of Pharmacy, Federal University of Pará, Belém 66075-110, PA, Brazil; suelem.farias@ics.ufpa.br (S.D.P.F.); regina1435vanessa@gmail.com (V.R.S.M.)

**Keywords:** fuclear factor NF-kappa-B, *Teucrium polium* L., monoterpenes, sesquiterpenes, verbenol, myrtenal, myrtenol

## Abstract

*Teucrium polium* L. is a plant with various claims of ethnobotanical use, primarily for inflammatory diseases. Chemical studies have already isolated different types of terpenes from the species, and studies have established its pharmacological potential. The present study evaluates the components of *T. polium* essential oil cultivated in the Algerian Saharan Atlas. GC-MS identified the major components as fenchone (31.25%), 3-carene (15.77%), cis-limonene oxide (9.77%), and myrcene (9.15%). In the in silico prediction, molecules with more than 1% abundance were selected. Regarding Lipinski’s rule, all molecules followed the rule. All molecules were found to be toxic in at least one model, with some molecules being non-genotoxic (6, 8, 10, 11, 12, 13) and others being non-mutagenic (5, 7, 9, 14). Three molecules were selected that showed the best results in pharmacokinetic and toxicity studies: the molecules that did not present carcinogenic potential (7—myrtenal; 9—myrtenol; 14—verbenol). The molecular target was established, and it seems that all three bound to the nuclear factor NF-kappa-B. Based on the docking and molecular dynamics results, these molecules have potential as anti-inflammatory and antitumor therapies, with further in vitro and in vivo studies needed to evaluate their activity and toxicity.

## 1. Introduction

*Teucrium polium* L. (Lamiaceae) is found in Europe, North Africa, and Asia. The following medicinal claims are attributed to it: treatment of inflammatory diseases, gastrointestinal disorders, diabetes, rheumatisms, indigestion, abdominal pain, colds, and urogenital diseases [1,2].

Chemical studies conducted on *T. polium* oil have identified compounds belonging to the following classes: sesquiterpenes (α- and τ-cadinols), (E)-β-caryophyllene and its oxide forms, neoclerodane diterpenoids, and monoterpenes. The proportions of these chemical constituents vary according to the collection site [2,3] and, possibly, factors such as the time of plant collection and the part used for oil extraction, among others.

The following compounds have already been identified in *T. polium* oil and listed as major components in at least one study: β-caryophyllene [4,5,6,7,8], germacrene D [5], limonene [5,9], p-cymene, 2,4-di-tert-butylphenol [9], α-pinene [6,10], α-thujene, terpinen-4-ol [10], ledol oxide (II), linalyl acetate, β-eudesmol [11], α-cardinol, caryophyllene oxide, epi-α-muurolol, cadalene, longiverbenone, carvacrol [6], 11-acetoxyeudesman-4-α-ol, α-bisabolol [7], β-pinene, α-muurolol, α-cadinol, α-muurolol, α-cardinol, α-cardinol [8], caryophyllene, γ-muurolene, cadinol, α-gurjunene, rosifoliol, 3-carene, γ-muurolene, α-phellandrene [12], carvacrol, torreyol [13], lycopersen, dodecane, 1,5-dimethyl decahydro naphthalene, tridecane [14], myrcene, menthofuran, ocimene, pulegone [15], β-eudesmol [16], β-pinene, limonene, α-phellandrene, linalool, terpinen-4-ol, γ- and δ-cadinenes, cedrol, cedrenol, and guaiol. In summary, more than 80 molecules have been identified in *T. polium* oils [17].

The essential oil of *T. polium*, with α-pinene, linalool, and caryophyllene oxide as its major components, demonstrates activity against Gram-positive bacteria (*Staphylococcus aureus* and *Staphylococcus epidermidis*) and Gram-negative bacteria [18]. Essential oils obtained from subspecies also show activity against *Acinetobacter baumannii* and *Staphylococcus aureus* [18].

*T. polium* is known for its antidiabetic effects through various mechanisms, such as increasing insulin secretion and levels, inducing the regeneration of pancreatic β-cells, reducing oxidative damage, promoting glucose uptake in muscle tissues, inhibiting α-amylase activity, and enhancing GLUT-4 translocation [2]. The antidiabetic effect has also been observed in male Wistar rats induced with diabetes by STZ injection (60 mg/kg, i.p.) and treated with *Teucrium polium* extract (100, 200, and 400 mg/kg) via daily gavage for 6 weeks. The results showed that the group treated with the extract exhibited reductions in glucose, triglycerides, and serum cholesterol, in addition to an attenuation of oxidative stress in aortic and cardiac tissues [19].

Due to its antimicrobial and antidiabetic potential, as well as its variation in chemical composition, it is crucial to identify the possible pharmacological markers of the species and their potential mechanisms of action, toxicity, and other aspects. In this context, in silico studies prove to be an important tool for predicting molecular structures and potential mechanisms of action of such compounds, as this type of study allows for the computational simulation of compounds from databases to predict various parameters such as physicochemical, pharmacokinetic, and toxicological properties [20].

This work is based on the analysis of the essential oil (EO) extracted from *T. polium*, with the major molecules selected for investigation related to physicochemical, pharmacokinetic, and toxicological predictions, biological activities, and potential targets of action. Subsequently, molecular modeling of the selected compounds is performed.

## 2. Materials and Methods

### 2.1. Chemical Studies

#### 2.1.1. Plant Material, and Extraction of the Essential Oil

The aerial parts of *T. polium* L. (Lamiaceae) were collected in April 2023 from Laghouat city (located in the south part of the Algerian Saharan Atlas); the GPS coordinates were 33°47′59″ N 2°51′54″ E. The plant material was taxonomically identified by the botanical survey, and its voucher specimen (LBAS Tp/04/23) was deposited in the Herbarium of the Laboratory of Biological and Agricultural Sciences, University of Amar Telidji, Laghouat, Algeria. After drying and grinding the plant, 100 g of powder was mixed with 1.5 L of distilled water in a round-bottomed flask and placed in a Clevenger-type apparatus for hydrodistillation. After 3 h, the essential oil was recuperated and stored in a sealed vial at 4 °C until analysis.

#### 2.1.2. Chromatographic Analysis

For the analysis of the essential oil, a Shimadzu GCMS QP 2010 ULTRA (Kyoto, Japan) with an RXİ-5MS capillary column (30 m × 0.25 mm inner diameter, film thickness 0.25 μm) was used. The percentage composition of the essential oil was written by calculating gas chromatography flame ionization detection (GC-FID) peaks.

The RXİ-5MS capillary column (30 m × 0.25 mm i.d., film thickness 0.25 µm) was used with helium as the carrier gas. The injector temperature was 250 °C, and the split flow was 1 mL/min. The GC oven temperature was kept at 40 °C for 3 min, programmed to 240 °C at a rate of 4 °C/min, and then kept constant at 240 °C for 53 min. For chemical component identification, Wiley and NIST electronic libraries were used [21,22]. For this study, service was purchased from Kastamonu University Central Research Laboratory “MERLAB” in Turkey.

### 2.2. In Silico Evaluation

The molecules were drawn using the Marvin JS online program (https://marvinjs-demo.chemaxon.com/latest/demo.html (accessed on 1 October 2024) and saved in the “Smiles” format for use on online servers (Appendix A), while, for the determination of physicochemical properties, the online server Home-ADMElab was used (https://admet.scbdd.com) [23]. The Lipinski’s rule of five or “Rule of Five” was considered [24]. For pharmacokinetic and toxicity predictions, the PreADMET program (version 2.0, Copyright © 2005–2017) was used, which considers pharmacokinetic properties (A—absorption; D—distribution; M—metabolism/biotransformation; E—excretion) and the evaluation of toxicity parameters (T—toxicity; PREADMET, 2020).

For the assessment of toxicity in marine organisms, the criteria used were as follows. For toxicity in algae [25], *Daphnia* sp. [26], medaka [27], and for [25], the mutagenicity risk was assessed by the Ames test with the following strains of *Salmonella typhimurium*: TA100-10RLI and TA 100-NA mutation in His G46e plasmid pKM101 without S9 and TA1535-10RLI and TA1535-NA mutation in His G46 [28].

The carcinogenic potential of the compounds was evaluated in rats and mice and referred to as (+) carcinogenic and (−) non-carcinogenic. To predict acute oral toxicity (lethal dose 50%-LD_50_), the online software PROTOX II was used [29], considering the classification from I to VI [30]. Adverse events that may occur with the use of the molecules were also evaluated.

### 2.3. Molecular Target and Docking

Based on the results obtained from in silico studies, particularly regarding carcinogenicity and mutagenicity, the molecules myrtenal, myrtenol, and verbenol were selected for docking. Initially, these molecules were submitted to the SuperPred Webserver [31], a validated server based on machine learning and similarity that utilizes the Smiles codes of promising molecules [32,33] (Appendix A). The server predicts molecular targets with potential interactions with the investigated ligands that may be related to anti-inflammatory activity and cancer.

The only target that showed relevance for the investigated biological activity was the nuclear factor NF-kappa-B p105, obtained from the protein data bank (PDB ID 1SVC), as the compounds with this target achieved scores for therapeutic activity interaction (≥ 90% binding probability and ≥90% prediction accuracy). Other targets, such as DNA-(apurinic or apyrimidinic site) lyase and the LSD1/CoREST complex, were not used because, despite their potential therapeutic activity, they showed a binding probability and a prediction accuracy below 90%.

Initially, the molecular structures of parthenolide, myrtenal, myrtenol, and verbenol were retrieved from the PubChem database and optimized using the DFT/B3LYP/cc-pVDZ quantum method with the Gaussian 09 program. The crystallographic structure of the nuclear factor NF-kappa-B p105 enzyme was obtained from the protein data bank (PDB ID: 1SVC) [34]. This PDB structure consisted of 364 amino acids, corresponding to residues 2 to 365 of the full 968-amino-acid sequence [35]. Among the 968 residues, the domain spanning amino acids 42 to 367, known as the Rel homology domain (RHD), binds to DNA at the major groove and is responsible for the transcriptional activity of the protein. Therefore, this region represents a potential binding site for small molecules aimed at inhibiting DNA transcription and was selected as the protein’s binding site, as proposed in the study [36].

Molecular docking was performed using the Molegro Virtual Docker (MVD) version 5.5 program [37]. The center of the sphere was defined with coordinates x: 40.37, y: 27.49, and z: 44.60, with a radius of 12 Å. The scoring function used was the MolDock Score. In addition, the inhibitor parthenolide [38] was included as a positive control in the study to enable a direct comparison of binding energy values and interactions with amino acid residues between the phytocompounds and this reference compound.

An analysis of intermolecular interactions was carried out using the Discovery Studio Visualizer (Dassault Systèmes BIOVIA, Discovery Studio Modeling Environment, version 2021, San Diego: Dassault Systèmes, 2021).

### 2.4. Molecular Dynamics

To gain further insights into the dynamic behavior and intermolecular interactions, the protein in its unbound form (apo) and in a complex with parthenolide (reference inhibitor), myrtenal, myrtenol, and verbenol was subjected to molecular dynamics (MD) simulations using the GPU-accelerated Amber22 software [39]. The restrained electrostatic potential (RESP) procedure was used to calculate the atomic charges of the ligands using the Gaussian 09 program at the HF/6-31G theory level [40]. The structures of the protein and the ligands were treated using the amber force field ff14SB and the general amber force field (GAFF), respectively [41,42].

The protonation states of the amino acid residues were calculated at pH 7.4 using the PDB2PQR server [43]. A TIP3P water box with a 12 Å radius was used to solvate the systems, and counterions were added to neutralize the system’s charges. To neutralize the systems and maintain a physiological concentration (0.15 M), Na^+^ and Cl^−^ ions were added [44].

Each solvated system was minimized in four stages: (i) ions and water molecules, (ii) hydrogen atoms, (iii) water molecules and hydrogen atoms, and (iv) the entire system. All steps were performed using 5000 steps with the steepest descent method and 5000 additional steps with the conjugate gradient algorithm. Subsequently, each system was heated for 200 ps to 300 K under a constant volume with positional restraints on the solute. An unrestrained equilibration step of 1 ns under a constant pressure was performed. Langevin dynamics was employed to control the temperature (300 K) with a collision frequency of 2 ps^–1^. The SHAKE algorithm [44] was used to restrain the bond lengths involving hydrogen atoms, while the particle mesh Ewald (PME) method [45] was employed to handle long-range electrostatic interactions. A 10 Å cutoff was applied for non-bonded interactions.

Finally, 200 ns of production was conducted without positional restraints at a constant temperature of 300 K. The pressure was controlled by a Berendsen barostat. The structural analysis of each system was performed by calculating the root mean square deviations (RMSD) and the root mean square fluctuations (RMSF) of the backbone atoms of the protein.

### 2.5. MM-GBSA Binding Free Energy Calculation

To estimate the binding free energy (ΔGbind) of the compounds parthenolide, myrtenal, myrtenol, and verbenol with the nuclear factor NF-kappa-B p105 protein, we used the MM-GBSA method implemented in the AmberTools23 [46]. The calculations utilized the final 10 ns (1000 frames) of the MD simulation trajectories. Established literature provides detailed descriptions of the MM-GBSA equations [47,48].

## 3. Results

### 3.1. Characterization of T. polium Essential Oil

Thirty-three chemical compounds were identified, representing 92.62% of the *T. polium* essential oil from the aerial parts (Table 1, Figure 1). Generally, the total amounts of monoterpene hydrocarbons in the essential oil were higher than in other groups. In the characterization of *T. polium* oil, 14 molecules were identified with concentrations of 1% or greater (Figure 1), with the major compounds being fenchone (31.25%), 3-carene (15.77%), limonene oxide, cis- (9.77%), and myrcene (9.15%). An additional 10 compounds were present with concentrations of 1% or greater.

In this study, terpenes were selected, including four monoterpenes (3-carene, Figure 2a; myrcene, Figure 2d; β-ocimene, Figure 2m, (E)-; verbenol, Figure 2n), five monoterpenoids (fenchone, Figure 2a; limonene oxide, cis-, Figure 2c; cis-pinocarveol, Figure 2e; myrtenal, Figure 2g; myrtenol, Figure 2i), three sesquiterpenes (germacrene D, Figure 2f; bicyclogermacrene, Figure 2h; δ-cadinene, Figure 2l), and two sesquiterpenoids (spathulenol, Figure 2j; (Z)-nerolidyl acetate, Figure 2k).

### 3.2. Predictions of Physicochemical, Pharmacokinetic, and Toxicity Aspects

No molecule violated Lipinski’s rule with adaptation; however, it is worth noting that all exhibited very low polar surface areas (0 to 26.3 Å) and reduced numbers of hydrogen bond acceptors and donors (Table 2).

An analysis of the pharmacokinetic parameters suggests that all molecules had moderate permeability in Caco-2 cells, moderate-to-high permeability in MDCK cells, and high intestinal absorption. Some molecules appeared to have a low potential for binding to plasma proteins and a moderate distribution to the central nervous system (CNS) (3 and 7), while others, despite their high plasma protein binding, seemed to have a high potential for distribution to the CNS (4, 6, 8, 11, 12, 13, and 14). All molecules underwent phase 1 metabolism by CYP3A4 and inhibited at least one CYP enzyme (Table 3).

All molecules were shown to be toxic to some marine organisms; however, the molecules that appeared to have no mutagenic potential (6, 8, 10, 11, 12, and 13) were carcinogenic to mice and rats (6, 8, 11, 12, and 13) or only to rats (10). On the other hand, the molecules that were not carcinogenic to any animal species (5, 7, 9, and 14) showed mutagenic potential (Table 4). Considering all the evaluated toxicities, it can be suggested that, despite their mutagenic potential, molecules 5, 7, 9, and 14 were the most promising.

Another aspect evaluated was the potential acute oral toxicity of the molecule, with the highest LD50 found for compound 11 (Class VI). Other molecules exhibited an LD50 greater than 2000 mg/kg (1, 2, 7, 8, 10, 12, 13, and 14). The possible side effects of these molecules were also assessed, with no events reported for 3, 5, 6, 12, and 13 (Table 5).

### 3.3. Predictions of Potential Molecular Targets of Compounds in T. polium Essential Oil

Based on the studies of the physicochemical predictions, pharmacokinetics, and toxicity, it can be suggested that the most promising molecules were 7, 9, and 14. Subsequently, targets with potential for biological activity related to cancer (nuclear factor NF-kappa-B p105 subunit) were identified with a correction and precision probability greater than 90%, and the PDB (protein data bank) code (1SVC) for docking was obtained through the online server, as shown in Table 6.

### 3.4. Docking Molecular Simulation

The compounds myrtenal, myrtenol, and verbenol, as well as the reference inhibitor parthenolide, were evaluated for their interactions with the residues of the NF-κB protein. These interactions are crucial for understanding the ligands’ affinity and specificity for the protein’s active site. Parthenolide exhibited hydrogen bonds with the residues Gly68, Ser66, and Pro65, along with extensive van der Waals interactions with several residues such as Gly116, Gly141, and Val115, reinforcing its role as an established NF-κB protein inhibitor (Figure 3).

In comparison, the compound myrtenal formed hydrogen bonds with the residues Arg57, Arg59, and Gly141, and alkyl-type interactions with Pro65 and Val115. This distribution of interactions suggests that myrtenal has a robust binding pattern similar to that of parthenolide, including critical interactions with Arg59 and Gly141 which may explain its higher stability observed in molecular dynamics analyses. Myrtenol interacted through hydrogen bonds with Tyr60 and Val61, while establishing pi–alkyl interactions with Arg59 and Val115 and alkyl interactions with Phe56 and His67. Although myrtenol exhibited multiple interactions at the active site, the combination of these interactions appeared to induce greater structural instability, as observed in the molecular dynamics results (Figure 4).

Verbenol, on the other hand, displayed hydrogen bonds with Pro65 and Gly68, as well as pi-alkyl interactions with Arg59 and alkyl interactions with Phe56, Val115, and Ile142. The interaction pattern of verbenol lies between that of myrtenal and myrtenol, highlighting its potential as a promising ligand, with a greater affinity for the active site than myrtenol but a lower dynamic stability compared to myrtenal (Figure 4).

### 3.5. Molecular Dynamics Result

The RMSD graph (Figure 5A) demonstrates the structural stability of the NF-κB protein in its unbound form (apo) and in complexes with the reference inhibitor parthenolide, as well as with the compounds myrtenal, myrtenol, and verbenol over 200 ns of simulations. It was observed that the complex with myrtenal presented the lowest average RMSD value (4.11 Å), indicating a superior dynamic stability compared to the other ligands, including parthenolide (6.76 Å). This result suggests that myrtenal may interact very efficiently with the protein’s active site, potentially on par with or better than the reference inhibitor. Myrtenol and verbenol exhibited average RMSD values of 5.44 Å and 7.29 Å, respectively, with myrtenol showing a greater structural instability, while verbenol displayed a behavior intermediate between that of myrtenol and that of parthenolide.

The RMSF graph (Figure 5B) corroborates these findings by revealing the protein’s residual fluctuations in different regions. In the binding site region (residues 16–26), myrtenal demonstrated lower fluctuations compared to the other ligands, indicating a greater local stabilization capability. Parthenolide, although effective, exhibited slightly higher residual fluctuations in this region, highlighting the potential of the proposed molecules to achieve competitive or even superior performance. Myrtenol, on the other hand, induced more pronounced fluctuations in various regions, which may indicate fewer specific interactions or the need for more significant conformational adjustments to fit into the active site. Verbenol, with fluctuations similar to those of parthenolide, displayed a promising dynamic profile.

These data are particularly significant, as they highlight the potential of myrtenal, myrtenol, and verbenol as viable alternatives to the reference inhibitor parthenolide. While parthenolide is widely recognized as an effective NF-κB protein modulator, the proposed compounds, especially myrtenal, demonstrate dynamic properties that make them promising for therapeutic applications. Myrtenal’s superior stability suggests that it could be explored as a highly competitive candidate, offering a foundation for chemical optimizations and further studies.

The RMSD graph (Figure 5A) illustrates the structural stability of the NF-κB protein in its unbound form (apo) and when complexed with parthenolide, myrtenal, myrtenol, and verbenol over 200 ns of simulations. The average RMSD values for the protein in the apo, parthenolide, myrtenal, myrtenol, and verbenol forms were 5.39 Å, 6.76 Å 4.11 Å, 5.44 Å, and 7.29 Å, respectively (Figure 5A). The myrtenal compound exhibited a greater stability and less fluctuations compared to myrtenol and verbenol, with a value close to that of the apo protein, indicating that this compound is dynamically more efficient in stabilizing the protein.

The RMSF graph (Figure 5B) illustrates the average residual fluctuations over time for each residue of the protein in its different forms. It was observed that the largest fluctuations were particularly pronounced in specific residues, especially between residues 32–37 and residues 246–253, corresponding to loop regions, which are more flexible. In the region where the ligand was accommodated, between residues 16–26, there was a lower fluctuation level in the complex with myrtenal, a phenomenon which is consistent with the RMSD data. Notably, the complex with myrtenol showed the highest fluctuations in several regions of the protein, corroborating the RMSD observation that this ligand induces a greater structural instability. The fluctuations observed in the complexes with myrtenal and verbenol were comparable to and smaller than those with myrtenol, suggesting that these ligands have a lesser impact on the protein’s dynamics.

The greater instability observed with myrtenol may be associated with a weaker or less specific binding to the active site or to the induction of larger conformational adjustments in the protein to accommodate the ligand. In contrast, the relatively stable behavior of the protein in complexes with myrtenal and verbenol suggests that these ligands are more compatible with the active site, resulting in smaller conformational fluctuations. These data are crucial for understanding the structure–function relationship and can guide future studies in the chemical modification of these ligands to enhance their efficacy and specificity.

### 3.6. MM-GBSA Binding Energies

The binding energies (ΔGbind) were calculated for the parthenolide-1SVC, myrtenal-1SVC, myrtenol-1SVC, and verbenol-1SVC complexes using the MM-GBSA method. The interaction energy components, including the van der Waals energies (ΔEvdw), electrostatic energies (ΔEele), polar solvation free energy (ΔGGB), and apolar solvation free energy (ΔGSA), were analyzed for each complex (Table 7).

The results show that the myrtenal-1SVC complex presented the most favorable binding energy (ΔGbind = −26.33 ± 3.57 kcal/mol), followed by verbenol-1SVC (ΔGbind = −22.14 ± 3.36 kcal/mol) and myrtenol-1SVC (ΔGbind = −17.64 ± 3.65 kcal/mol). These values indicate that myrtenal forms the most stable complex with the 1SVC protein, a finding which is consistent with the lower conformational fluctuations observed in the RMSD and RMSF data, suggesting a strong interaction of this compound with the protein’s interaction site.

Comparatively, parthenolide (the reference inhibitor) exhibited a binding energy of ΔGbind = −15.47 ± 2.25 kcal/mol, which is superior in stability compared to that of myrtenol but inferior to that of myrtenal and verbenol. This difference suggests that, while parthenolide interacts effectively with the protein, the compounds myrtenal and verbenol exhibited stronger and more stable binding affinities.

The analysis of the energy components revealed that, in all complexes, electrostatic energy (ΔEele) played a predominant role in stabilizing ligand–protein interactions, especially in the case of myrtenal-1SVC, which showed the most negative ΔEele value (−98.53 ± 8.44 kcal/mol). However, this strong electrostatic contribution was partially counterbalanced by polar solvation energy (ΔGGB), which was higher for myrtenal, indicating that the electrostatic interactions were strongly solvated.

The MM-GBSA analysis results reinforce the observations from the RMSD and RMSF analyses. Myrtenal, which showed the most negative free binding energy (−26.33 ± 3.57 kcal/mol), also induced the lowest structural fluctuations, suggesting a combination of strong interactions and dynamic conformational fit. Verbenol, with a free binding energy of −22.14 ± 3.36 kcal/mol, provided a better conformational stability than myrtenol, as observed in the RMSD and RMSF analyses, indicating that, while its binding affinity is lower than that of myrtenal, it still presents a good potential for protein modulation.

These results indicate that both myrtenal and verbenol stand out as compounds with a stronger inhibition potential relative to NF-κB compared to parthenolide, with myrtenal showing the highest potential, followed by verbenol, while myrtenol exhibited a less stable interaction profile.

## 4. Discussion

The essential oil obtained from *T. polium* was subjected to GC-MS analysis, revealing the major constituents as fenchone (31.25%), 3-carene (15.77%), limonene oxide, cis- (9.77%), and myrcene (9.15%). When comparing these results to other studies, it is observed that other metabolites such as β-caryophyllene [3], limonene [10], ledene oxide II [11], α-cardinol [51], carvacrol [6], and β-pinene were the major constituents [52]. Studies on the environmental impact on the composition of *T. polium* oil are still scarce; however, it is known that factors such as altitude, water availability, macro and micronutrients in the soil, relative air temperature, and soil pH directly affect the chemical profile of plants [53].

Myrcene was reported in previous studies as the major component of the essential oil of *T. polium* [54,55,56,57,58]. Myrcene was found to be the major compound in our study, too. However, the main constituents of the essential oils of the aerial parts were oxygenated monoterpenes and monoterpene hydrocarbons, findings which are in good agreement with the previous reports [54,59,60,61,62].

On the other hand, germacrene D was detected as a major compound in the essential oil of *T. polium* samples from different regions [58,60,63,64,65]. Similarly, germacrene D was detected as the main compound in our study. While fenchone, 3-carene, limonene oxide, and cis- were found to be the main compounds in our study, they were minor or absent in essential oils of *Teucrium* [10,63,66]. Therefore, environmental factors, the plant part used in the extraction process, and the collection time can influence the chemical composition of the essential oil.

All selected molecules adhered to Lipinski’s rule and appeared to exhibit high intestinal absorption. However, only molecules 2, 4, 5, 6, 8–14 were distributed to the CNS. Adhering to Lipinski’s rule is crucial for drug candidates, as it indicates that the drug will be well absorbed in the gastrointestinal tract and adequately distributed throughout the body, allowing for oral administration [24,32,67].

All molecules seemed to be metabolized by CYP3A4, but they inhibited CYP and, sometimes, more than one CYP. Molecules that inhibit CYP can interfere with the metabolism of other drugs, necessitating dose adjustments. Another evaluated parameter was toxicity, with 8, 10, 11, 12, 13 not being mutagenic, while 7, 9, and 14 were not carcinogenic. Unfortunately, no compound was devoid of toxicity; however, all compounds had an LD50 > 1400 mg/kg. Therefore, repeated-dose toxicity studies, in vivo genotoxicity, and in vivo carcinogenicity studies are important for understanding toxic effects and potential mechanisms.

After analyzing the pharmacokinetic studies and toxicities, molecules without carcinogenic potential were selected (7—myrtenal; 9—myrtenol; 14—verbenol). Myrtenal exhibited antihyperglycemic effects, reducing blood glucose levels and hemoglobin A1C and aiding in weight recovery [68]. Derivatives of myrtenal have shown activity against various cell lines [68,69,70,71].

Other activities related to myrtenal derivatives include anxiolytic [72], antiviral [72], antifungal [73], and analgesic [74]. Another selected molecule was myrtenol, which inhibits biofilm formation and virulence in the drug-resistant *Acinetobacter baumannii*. Myrtenol improves the susceptibility of BP-AB to the antibiotic’s amikacin, piperacillin/tazobactam, cefoperazone/sulbactam, and ceftazidime. This molecule regulates the expression of biofilm-associated genes in the BP-AB strain, and qPCR analysis has been shown to reduce the expression levels of bfmR, bap, csuA/B, and ompA in groups D, E, and F compared to groups A, B, and C. A non-significant reduction in bfmR, bap, csuA/B, and ompA levels has also been found in groups A, B, and C. The genes bfmR, bap, csuA/B, and ompA are key regulators of the transition from biofilm formation to maturation in the BP-AB strain [75]. Myrtenol protects against myocardial ischemia–reperfusion injury through antioxidant and anti-apoptotic mechanisms [76], while verbenol exhibits anti-ischemic and anti-inflammatory properties [77].

To identify the potential target involved in the biological activity of myrtenal, myrtenol, and verbenol, prediction studies have been conducted, suggesting that all three bind to the nuclear factor NF-kappa-B, a family of transcription factors involved in inflammation, immunity, cell proliferation, differentiation, and survival [78]. In recent years, the presence and activation of the nuclear factor NF-kappa-B in different types of cancer have been highlighted, as well as the importance of developing inhibitors that act directly on the nuclear factor NF-kappa-B [79]. The possibility of therapeutically targeting this factor allows for a significant advance in tumor destruction during treatment, thereby enhancing antitumor activity [80].

It is worth highlighting the medicinal importance of *Teucrium* species, which have been used since ancient times in the Mediterranean region for treating gastrointestinal issues and maintaining healthy endocrine gland functions, as well as treating malaria and severe dermatological disorders. However, studies evaluating their activity are scarce. Evaluations of the essential oils and ethanolic extracts of *Teucrium polium* and *Teucrium parviflorum* have shown that the extracts exhibit antioxidant, anti-butyrylcholinesterase, anti-tyrosinase, and anti-urease activities through in vitro and in silico assays [81]. It is noteworthy that *T. polium* oil demonstrates a moderate antioxidant potential [82].

An in vivo study with the ethanolic extract of *T. polium* demonstrated the plant’s anti-inflammatory potential at concentrations of 50 mg/kg, 100 mg/kg, and 150 mg/kg, leading to a reduction in paw edema in rats [83]. When correlating this result with prediction studies, the regulation of NF-κB activity is crucial to prevent chronic inflammation, meaning that substances with anti-inflammatory activity can suppress NF-κB activation or interfere with its translocation to the nucleus, reducing the expression of inflammatory genes [79]. In addition to its involvement in the inflammatory process, NF-kappa-B (NF-κB) is involved in cell proliferation, apoptosis (programmed cell death), stress response, and other aspects relevant to cancer development and progression [80].

It should be noted that the chronic inflammation process favors mutations, uncontrolled cell proliferation, and resistance to apoptosis, all of which are processes that can facilitate carcinogenesis [84]. Furthermore, NF-κB induces the production of the vascular endothelial growth factor (VEGF) and regulates molecules involved in cell mobility and tissue invasion, such as matrix metalloproteinases (MMPs) [85,86]. Considering this, it can be suggested that these molecules hold promise as antitumor and anti-inflammatory agents, and in vitro and in vivo studies are necessary to determine the best therapeutic use for these molecules.

The results obtained in the present study demonstrate that the compounds myrtenal, myrtenol, and verbenol exhibit significant interactions with the active site residues of the NF-κB protein, with binding patterns comparable to those of the reference inhibitor parthenolide. These interactions are particularly relevant, considering that the NF-κB protein plays a central role in regulating inflammatory responses and is associated with various pathologies such as cancer, autoimmune diseases, and chronic inflammation [87]. The presence of strong interactions with critical residues in the active site suggests that the tested compounds have the potential to modulate NF-κB activity, influencing the transcription of genes involved in inflammation and disease development.

Previous docking studies of the DNA-NF-κB protein have revealed that the compounds myrtenal, myrtenol, and verbenol bind to the same pocket where the DNA interacts, a crucial point for stabilizing the ligand–protein complex. This binding at the DNA interaction site is fundamental, as it blocks the nuclear translocation of the NF-κB transcription factor, preventing its activation and subsequent expression of inflammatory genes. These findings are consistent with results described in the literature, indicating that compounds with interaction patterns similar to those of parthenolide have a greater potential as NF-κB inhibitors [36]. The similarity of the interactions with key residues at the protein’s active site and the ability to block its nuclear translocation reinforce the idea that the studied compounds may represent promising strategies for developing new therapeutic agents targeting inflammatory diseases and other conditions associated with exacerbated NF-κB activation.

The results highlight myrtenal as the most promising compound among those tested, with interactions similar to those of parthenolide, including critical residues at the active site of the NF-κB protein. This underscores its potential as an effective and stable ligand, potentially contributing to the functional inhibition of NF-κB.

## 5. Conclusions

Based on the results of the molecular docking, molecular dynamics, and free energy calculations, this study suggests that the most promising compounds for modulating the NF-κB protein are myrtenal and verbenol, with myrtenal standing out due to its high stability and binding affinity to the protein’s active site. Myrtenal exhibited the most negative binding energy value (ΔGbind = −26.33 ± 3.57 kcal/mol), indicating a strong and stable interaction with the protein which was corroborated by molecular dynamics simulations revealing lower structural fluctuations (RMSD and RMSF) compared to other compounds, including parthenolide, the reference inhibitor.

Furthermore, the docking data and molecular interaction analysis indicated that myrtenal and verbenol exhibit robust binding patterns, with critical interactions with key residues of the NF-κB protein. These compounds demonstrated not only significant inhibition potential against NF-κB, but also lower conformational instability, suggesting they are viable alternatives for the development of anti-inflammatory and antitumor therapies.

Based on these results, additional chemical studies will be conducted to isolate the priority molecules from the essential oil of *T. polium*. After isolation, in vitro assays will be planned, including evaluations of cytotoxicity, genotoxicity, mutagenicity, and mechanisms of cell death. The active compound with the lowest toxic potential will be subjected to studies to assess its mechanisms of action, followed by structural modifications to optimize its inhibitory potential and reduce toxicity.

The final phase of pharmacological studies will involve in vivo testing (toxicity and activity) to establish dose–response correlations. If the pharmacological potential is confirmed, it will be possible to move forward with product development. In summary, the essential oil of *T. polium*, due to its composition, shows great promise as an anti-inflammatory and antitumor agent, with the potential for new treatments based on the compounds myrtenal and erbenol.

## Figures and Tables

**Figure 1 cimb-47-00048-f001:**
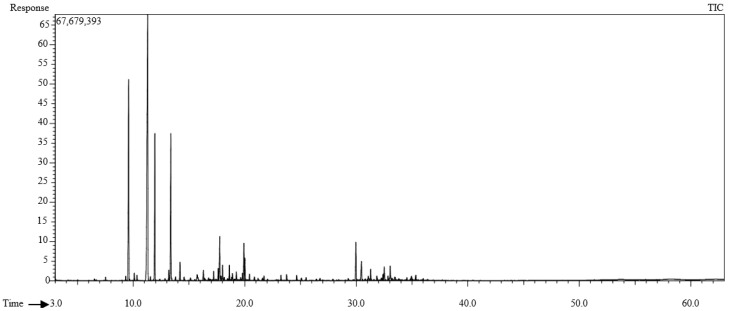
Gas chromatography flame ionization detector (GC-FID) profile of the essential oil of *Teucrium polium*.

**Figure 2 cimb-47-00048-f002:**
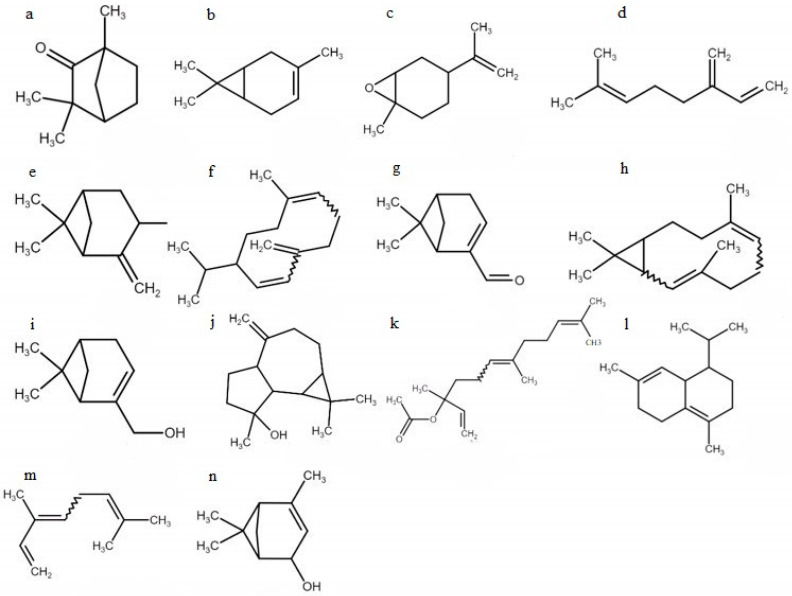
Molecules found in *T. polium* essential oil: (**a**)—fenchone; (**b**)—3-carene; (**c**)—limonene oxide, cis-; (**d**)—myrcene; (**e**)—cis-pinocarveol; (**f**)—germacrene D; (**g**)—myrtenal; (**h**)—bicyclogermacrene; (**i**)—myrtenol; (**j**)—spathulenol; (**k**)—(Z)-nerolidyl acetate; (**l**)—δ-cadinene; (**m**)—β-ocimene, (E)-; (**n**)—verbenol.

**Figure 3 cimb-47-00048-f003:**
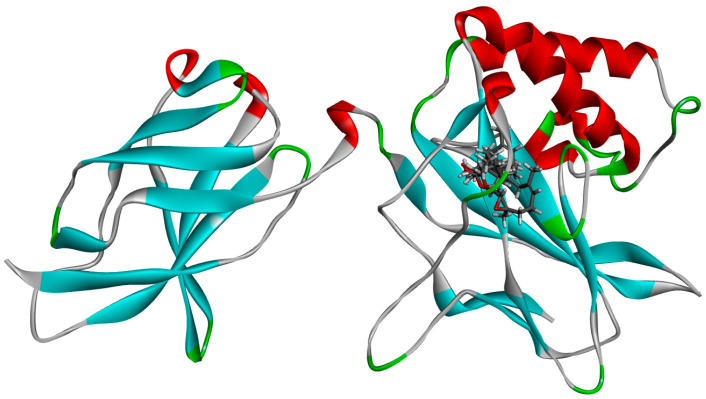
Illustration of the compounds docked on the active site of the NF-κB protein.

**Figure 4 cimb-47-00048-f004:**
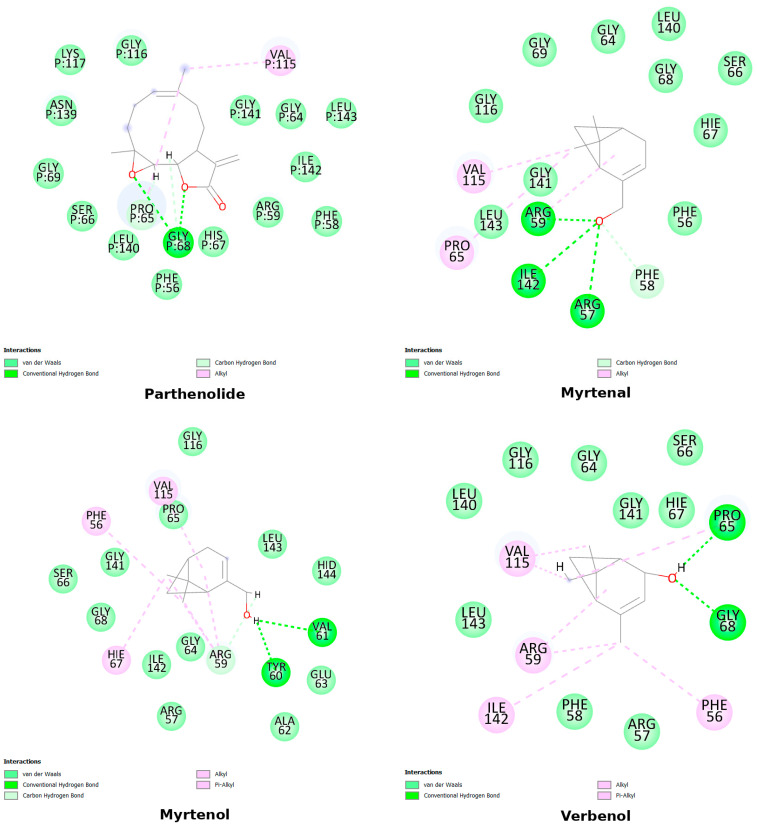
Representation of the 2D interactions of the molecules parthenolide, myrtenal, myrtenol, verbenol, and the protein nuclear factor NF-kappa-B. Image generated with Discovery Studio 3.5 Visualizer.

**Figure 5 cimb-47-00048-f005:**
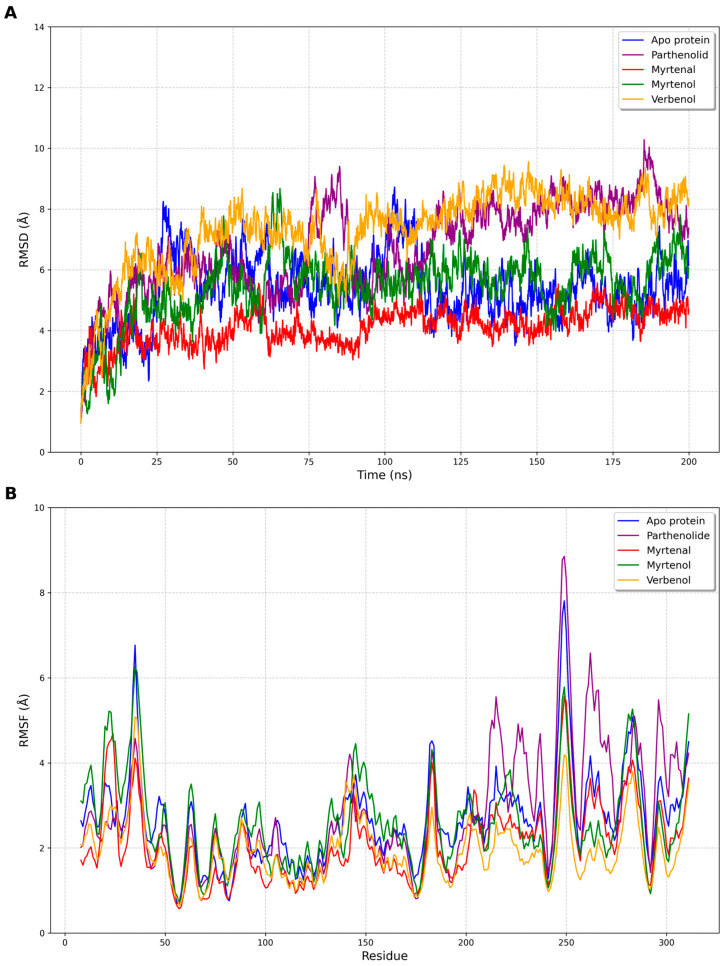
Chart of RMSD (**A**) and RSMF (**B**) of the apo form of the protein nuclear factor NF-kappa-B and complexed with parthenolide, myrtenal, myrtenol, and verbenol.

**Table 1 cimb-47-00048-t001:** Essential oil composition of the aerial parts of *Teucrium polium*.

	RRI	References	Compounds	RA (%)
1	946	939–957	Camphene	0.40
2	953	937–953	Verbenene	0.26
3	1008	997–1027	3-Carene	15.77
4	1009	990–1009	α-Phellandrene	0.75
5	1055	1059–1087	Fenchone	31.25
6	1064	1027–1050	β-Ocimene, (E)-	1.02
7	1089	1089	p-Cymene	0.65
8	1122	1106–1134	α-Campholenal	0.59
9	1132	1122–1144	Limonene oxide, cis-	9.77
10	1140	1140–1175	Myrcene	9.15
11	1146	1146	Verbenol	1.02
12	1150	1110–1150	δ-2-Carene	0.72
13	1160	1121–1158	Pinocarvone	0.91
14	1162	1147–1176	Linalool oxide	0.64
15	1165	1134–1165	cis-Verbenol	0.36
16	1169	1122–1169	3-Carene	0.80
17	1182	1182	cis-Pinocarveol	2.92
18	1186	1159–1191	α-Terpineol	0.46
19	1194	1169–1200	Myrtenol	1.47
20	1195	1171–1206	Myrtenal	2.31
21	1204	1190–1224	Verbenone	0.38
22	1235	1206–1235	Carvone	0.28
23	1254	1259–1284	Bornyl acetate	0.31
24	1270	1270–1302	Terpinen-4-ol acetate	0.54
25	1290	1290–1316	Myrtenyl acetate	0.70
26	1484	1458–1491	Germacrene D	2.56
27	1500	1474–1501	Bicyclogermacrene	1.56
28	1521	1508–1539	δ-Cadinene	1.18
29	1577	1562–1590	Spathulenol	1.47
30	1640	1610–1650	α-Muurolol, epi-	0.43
31	1649	1649–1686	α-Bisabolol	0.34
32	1654	1619–1662	α-Cadinol	0.35
33	1677	1676	(Z)-Nerolidyl acetate	1.30
Grouped compounds (%)				
Monoterpene hydrocarbons				43.15
Oxygenated monoterpenes				43.74
Sesquiterpenes hydrocarbons				5.73
Total identified compounds (%)				92.62

RRI: Relative retention indices, RA (≥0.25): relative area (peak area relative to the total peak area) [49,50].

**Table 2 cimb-47-00048-t002:** Prediction of physicochemical properties.

Molecules	MM	LogP	TPSA	nHBA	nHBD
1	152.237	2.402	17.07	1	0
2	136.238	2.999	0.00	0	0
3	152.237	2.520	12.53	1	0
4	136.238	3.475	0.00	0	0
5	152.237	1.970	20.23	1	1
6	204.357	4.891	0.00	0	0
7	150.221	2.178	17.07	1	0
8	204.357	4.725	0.00	0	0
9	152.237	1.971	20.23	1	1
10	220.356	3.386	20.23	1	1
11	264.409	4.967	26.30	2	0
12	204.357	4.725	0.00	0	0
13	136.238	3.475	0.00	0	0
14	152.237	1.970	20.23	1	1

Lipinski’s rule: LogP—oil–water partition coefficient ≤ 5; TPSA: topological polar surface area ≤ 140 Å; nHBA: number of hydrogen bond acceptors ≤ 10; nHBD: number of hydrogen bond donor groups ≤ 5; MM—molecular mass ≤ 500D [24]. 1—fenchone; 2—3-carene; 3—limonene oxide, cis-; 4—myrcene; 5—cis-pinocarveol; 6—germacrene D; 7—myrtenal; 8—bicyclogermacrene; 9—myrtenol; 10—spathulenol; 11—(Z)-nerolidyl acetate; 12—δ-cadinene; 13—β-ocimene, (E)-; 14—verbenol.

**Table 3 cimb-47-00048-t003:** Prediction of pharmacokinetic properties.

	Absorption		Distribution			Metabolism	
Molecules	MDCK	Caco 2	HIA	PP	BBB	CYP Inibition	CYP Phase 1
1	M	M	H	H	M	2C9, 3A4	3A4
2	H	M	H	H	H	2C9	3A4
3	H	M	H	L	M	2C9, 3A4	W 3A4
4	H	M	H	H	H	2C9, 3A4	3A4
5	M	M	H	L	H	2C9, 3A4	W 3A4
6	M	M	H	H	H	2C9, 2C19	3A4
7	H	M	H	L	M	2C9	W 3A4
8	M	M	H	H	H	2C9	3A4
9	H	M	H	L	H	2C9	W 3A4
10	H	M	H	L	H	2C9, 3A4	3A4
11	M	M	H	H	H	2C19, 2C9, 3A4	3A4
12	M	M	H	H	H	2C19, 2C9	3A4
13	M	M	H	H	H	2C19, 2C9	3A4
14	H	M	H	H	H	2C9	W 3A4

BBB: blood–brain barrier; CYP: cytochrome P450; HIA: human intestinal absorption; W: weakly; H: high; L: low; M: medium; 1—fenchone; 2—3-carene; 3—limonene oxide, cis-; 4—myrcene; 5—cis-pinocarveol; 6—germacrene D; 7—myrtenal; 8—bicyclogermacrene; 9—myrtenol; 10—spathulenol; 11—(Z)-nerolidyl acetate; 12—δ-cadinene; 13—β-ocimene, (E)-; 14—verbenol.

**Table 4 cimb-47-00048-t004:** Prediction of toxicity.

Molecules	Alga	Daphnia	Medaka Fish	Minnow Fish	Ames	Carcino Rato/Cam *
1	T	NT	VT	VT	TA1535_10RLI	N/P
2	T	NT	VT	VT	TA100_10RLI	N/P
3	T	NT	VT	VT	TA1535_10RL; 100_10RLI; 1535_NA	P/P
4	T	T	VT	VT	TA1535_NA	P/N
5	T	NT	VT	VT	TA100_10RLI; 1535_NA	N/N
6	T	T	VT	VT	N	P/P
7	T	NT	VT	VT	TA1535_10RLI; 100_10RLI	N/N
8	T	T	VT	VT	N	P/P
9	T	NT	VT	VT	TA1535_10RLI; 100_10RLI	N/N
10	T	T	VT	VT	N	P/N
11	T	T	VT	VT	N	P/P
12	T	T	VT	VT	N	P/P
13	T	T	VT	VT	N	P/P
14	T	NT	VT	VT	TA1535_10RLI; TA100_10RLI	N/N

T: toxic; NT: non-toxic; N: negative; P: positive. Parameters: algae—<1 mg/L toxic; >1 mg/L non-toxic [25]; daphnia test: < 0.22 µg/mL toxic; > 0.22 µg/mL—non-toxic [26]; test on medaka and minnow fish: < 1 mg/L—very toxic; 1–10 mg/L—toxic; 10–100 mg/L—harmful and > 100 mg/L—extremely toxic [27], carcino rat/mice * = carcinogenicity in rat/mice. T—toxic, NT—non-toxic, VT—very toxic, N—negative, P—positive. 1—fenchone; 2—3-carene; 3—limonene oxide, cis-; 4—myrcene; 5—cis-pinocarveol; 6—germacrene D; 7—myrtenal; 8—bicyclogermacrene; 9—myrtenol; 10—spathulenol; 11—(Z)-nerolidyl acetate; 12—δ-cadinene; 13—β-ocimene, (E)-; 14—verbenol.

**Table 5 cimb-47-00048-t005:** Prediction of oral toxicity.

Molecules	LD50 (mg/kg)	Toxicity Class	Side Effects
1	3087	V	I
2	2799	V	I/T
3	1447	IV	-
4	2561	V	I/T
5	1971	IV	-
6	1471	IV	-
7	2448	V	I
8	2766	V	I/T/M
9	1736	IV	I
10	3278	V	I/T
11	5923	VI	T
12	2090	V	-
13	2652	V	-
14	2280	V	I

LD50—lethal dose 50%. I—irritant, T—tumorigenic, M—mutagenicity. Category I: 1 < LD50 ≤ 5 mg/kg—extremely toxic; category II: 5 < LD50 ≤ 50 mg/kg—highly toxic; category III: 50 < LD50 ≤ 300 mg/kg—moderately toxic; category IV: 300 < LD50 ≤ 2000 mg/kg—low toxicity; category V: 2000 < LD50 ≤ 5000 unlikely to cause acute damage; category VI: DL50 > 5000 no damage. Source: [30] 1—fenchone; 2—3-carene; 3—limonene oxide, cis-; 4—myrcene; 5—cis-pinocarveol; 6—germacrene D; 7—myrtenal; 8—bicyclogermacrene; 9—myrtenol; 10—spathulenol; 11—(Z)-nerolidyl acetate; 12—δ-cadinene; 13—β-ocimene, (E)-; 14—verbenol.

**Table 6 cimb-47-00048-t006:** Molecular target assessment.

Molecules	Probability	Prediction Accuracy	Target Name	PDB
7	91.76%	96.09%	NF-kappa-B	1SVC
9	96.52%	96.09%	NF-kappa-B	1SVC
14	92.39%	96.09%	NF-kappa-B	1SVC

PDB: protein data bank; NF-kappa-B: nuclear factor NF-kappa-B p105 subunit; 7—myrtenal; 9—myrtenol; 14—verbenol.

**Table 7 cimb-47-00048-t007:** Binding energies and their components calculated by MM-GBSA (in kcal/mol).

Complex	ΔEvdw	ΔEele	ΔG_GB_	∆G_SA_	ΔG_bind_
Parthenolide	−24.24 ± 2.72	0.10 ± 3.25	−24.14 ± 5.15	8.67 ± 3.65	−15.47 ± 2.25
Myrtenal-1SVC	−8.92 ± 2.99	−98.53 ± 8.44	81.12 ± 5.85	−107.46 ± 7.33	− 26.33 ± 3.57
Myrtenol-1SVC	−5.59 ± 2.89	−58.77 ± 8.62	40.73 ± 5.94	−58.37 ± 7.47	− 17.64 ± 3.65
Verbenol-1SVC	−17.01 ± 2.59	−81.45 ± 8.51	76.32 ± 7.24	−98.46 ± 8.28	− 22.14 ± 3.36

## Data Availability

Data is contained within the article and Appendix A.

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
