# Peer review of "Prediction of the Binding to the Nuclear Factor NF-Kappa-B by Constituents from Teucrium polium L. Essential Oil"

_cimb, 2025, doi:10.3390/cimb47010048_

Round 1
Reviewer 1 Report
Comments and Suggestions for Authors
In the manuscript” Prediction of the binding to Nuclear Factor NF-kappa-B of constituents from Teucrium polium L. essential oil” (cimb-3345630) the authors analysed the essential oil (EO) extracted from T. polium, with the major molecules selected for investigations related to physicochemical, pharmacokinetic, toxicological predictions, biological activities, and potential targets of action.
The manuscript is interesting, however it should be improved.
I have a few minor concerns that can be easily addressed:
1) In the Table 1 there is Referencesa,b, what does it mean, what mean the symbol a and b?
2) The conclusions should be improved indicating what results from the studies and only one or two sentences of future studies.
3) In the text in lines 105-110 the Figure number 2 should be indicated and then markings of compounds (you cannot write it as individual Figures for each compounds). There are missing marks of compounds in the caption to that Figure 2.
4) The Table 3 should be included on the same page (not divided).
5) It should be better for reads to see docked selected compounds into the structure of NF-kappa-B so in addition to Figure 3 additional Figure with compounds in 3D structure of NF-kappa-B should be included.
6) There is no supplementary which were mentioned in the text.
7) On what basis was the NF-kappa-B protein selected as a target?
8) There is no in discussion any description of interactions in the binding site of target protein. What are important amino acids crucial for binding compounds in NF-kappa-B?
9) The manuscript should carefully checked for any text mistakes.
Reviewer 2 Report
Comments and Suggestions for Authors
The authors report the physicochemical and pharmaco-toxicological properties of selected Teucrium polium L. essential oil components and molecular docking simulations.
The topic of this work is interesting, and the authors tried to approach it in a complex manner. However, I have some concerns that authors should address:
* The authors should explain why they chose the Nuclear Factor NF-kappa-B p105 subunit as an inflammation and cancer-related target.
* What is the relevance of the preparation and chromatographic analysis of vegetable oil? Why was more than literature documentation (cited in the Discussion section) needed as a starting point for in silico analysis?
* Have the authors developed a new extraction method (no comparison with the literature) and a new chromatographic separation and quantification method (in terms of method, working conditions, column, etc.)?
* The authors should also approach molecular docking with a known drug ligand for NF-κB. They should compare and discuss the binding energy values ​​and amino acid residues bound to the phytocompounds with reference data.
Round 2
Reviewer 1 Report
Comments and Suggestions for Authors
The manuscript was corrected according suggestion however there isn't still any supplementary materials.
Reviewer 2 Report
Comments and Suggestions for Authors
The authors responded to all my comments in the Author Response File but prepared the revision hastily. They did not include in the manuscript the justifications for the choice of NF-kappa-B p105, although they provided an extensive response in their reply to the reviewer. Lines 564-566 are in Portuguese.
Round 3
Reviewer 2 Report
Comments and Suggestions for Authors
The authors addressed all comments appropriately and made corresponding changes, thus significantly improving their manuscript.